# Partner’s Perceived Social Support Influences Their Spouse’s Inflammation: An Actor–Partner Analysis

**DOI:** 10.3390/ijerph19020799

**Published:** 2022-01-12

**Authors:** Joshua D. Landvatter, Bert N. Uchino, Timothy W. Smith, Jos A. Bosch

**Affiliations:** Department of Psychology and Health Psychology Program, University of Utah, Salt Lake City, UT 84112, USA; Bert.Uchino@psych.utah.edu (B.N.U.); Tim.Smith@psych.utah.edu (T.W.S.); J.A.Bosch@uva.nl (J.A.B.)

**Keywords:** social support, dyad, inflammation, interpersonal, actor–partner

## Abstract

Social support has been linked to lower cardiovascular morbidity and mortality. However, most studies have examined perceived support as an intrapersonal construct. A dyadic approach to social support highlights how interdependence between individuals within relationships, including partner perceptions and interactions, can influence one’s health. This study’s overall purpose was to test actor–partner models linking perceived social support to inflammation. Ninety-four cisgender married couples completed perceived support measures and had their blood drawn for CRP and IL-6 to produce an overall inflammatory index. The primary results indicate that only a partner’s level of perceived support was related to lower inflammation in their spouse. Our sample size, although moderate for inflammatory studies, was probably not large enough to detect actor influences. These data highlight the importance of taking a dyadic perspective on modeling perceived support and its potential mechanism.

## 1. Introduction

The quality of one’s social relationships is reliably related to physical health outcomes [1,2,3]. In perhaps the most compelling evidence to date, Holt-Lunstad, Smith, and Layton (2010) conducted a meta-analysis of 148 studies comprising over 308,000 participants. They found evidence that lower social support was related to a nearly 50% higher risk for future mortality [4]. Barth and colleagues [5] similarly found strong evidence linking social support to lower cardiovascular disease risk, which remains the leading cause of death in the United States and other industrialized countries [6].

Much of the prior work has focused on the biological mechanisms underlying the link between social support and health [7]. Of these biological mechanisms, inflammation has emerged as one of the more compelling pathways, considering its mechanistic links to cardiovascular disease morbidity and mortality [8,9,10]. A recent meta-analysis reported a small effect size linking social support to inflammatory cytokines [11]. Given the links between social support and broad-based mortality [4,7], an inflammatory pathway might explain part of its ties to health.

An important limitation in the literature is that all studies to date appear to examine if a person’s level of social support predicts their own inflammatory outcomes [12,13,14]. This intra-individual perspective highlights models in which social support processes perceived by the individual might influence their health and assumes that these perceptions, in part, reflect past support exchanges. However, research on invisible support and early life determinants of perceived support makes it clear that there are interpersonal determinants of support that have implications for its conceptualization [15,16]. Such interpersonal determinants of social support can be tested using actor–partner models, highlighting the interdependence between individuals within relationships [17]. These models examine if one’s own characteristics or those of a partner predict outcomes. In this case, the question is whether one’s own level of perceived support (actor influences) or a partner’s level of perceived support (partner influences) predicts inflammatory outcomes.

There is strong evidence that close social network members influence inflammation [18]. In a study by Donoho and colleagues, 2013, marital quality was seen as an important predictor of inflammation on the inflammatory biomarkers of C-reactive protein and interleukin-6 [19]. However, only limited work has examined dyadic influences on biomarkers to date, revealing links with cortisol, cardiovascular reactivity, and inflammation. Coregulation of actor–partner cortisol levels has been shown between couples and their mood states, with marital satisfaction acting as a possible buffer to the partner’s negative mood or stress state [20,21]. Furthermore, dyadic effects on cardiovascular reactivity measures have been observed [22,23,24,25,26,27], including partner influences of trait hostility and anger [28]. A review examining the immune system, marriage, and divorce also found that partners influence each other’s mood and health behaviors, producing both direct and indirect downstream effects on the immune system [29]. This evidence suggests that one’s perceived social support may influence inflammatory biomarkers in their partner.

When conceptualized as a dyadic or interpersonal process, there are several specific reasons to expect partner influences of perceived support on health. For example, being married to a person high in perceived support might simply be less stressful compared to being married to someone low in support [18,30]. Given that the partner has adequate support, they would have the resources needed to attend to their partner’s socio-emotional needs [31,32]. Additionally, perceived support is related to received support. Therefore, if a partner has higher levels of perceived support, this may be reflective of more responsive reciprocity in support processes [33]. Due to the reciprocal nature of dyadic relationships, one’s perception of support can ultimately affect partner responsiveness and self-disclosure and act as a buffer against stress or negative moods [20,21,24]. These processes (e.g., support reciprocity) between an actor and their partner may cushion the effects of stressors and consequent health issues [34,35].

This study’s main goal was to investigate actor–partner models linking perceived support to inflammation and explore the interpersonal components of perceived support. This interpersonal component “fills in the gaps” of prior work by highlighting the interpersonal context of dyads that one’s own perception of support may not reflect. These questions were explored in married couples, providing an ideal interpersonal context due to the importance of marriage in adulthood [36]. Based on prior work, it was predicted that higher actor levels of perceived support would be related to lower inflammation. Due to the interpersonal processes associated with support [16], it was also predicted that an increase in a partner’s level of support would be related to lower levels of inflammation in the spouse.

## 2. Materials and Methods

### 2.1. Participants

All methods and procedures implemented in this study were pre-approved by the Institutional Review Board (IRB_00033677) of the University of Utah. Ninety-four relatively healthy married couples were recruited for this study. We included a wide age range (i.e., 42 to 78 years old, M_age_ = 56.2, SD = 7.30) to increase variability in inflammatory measures. Most participants were white (94.6%) and college educated (70.3%) and had an income of over USD 40,000 per year (87.6%). Because many middle-aged and older adults are on health-related medications, we only excluded individuals who (a) were undergoing strong immunosuppressive treatment (e.g., corticosteroid therapy) and/or (b) had cancer or HIV due to concerns about potential effects of treatment on inflammatory outcomes. Medication use was coded dichotomously (yes/no) based on its specific biological function (e.g., Lipitor as a statin). The presence of an acute illness that could influence inflammatory indices was determined by CRP values (>10). No participants had a CRP value over or close to 10. Additionally, participants were also screened for conditions related to cardiovascular disease, such as diabetes.

### 2.2. Procedure

To control for diurnal variations in inflammation, all eligible participants were scheduled together for an appointment at the University of Utah in a time window between 9 a.m. and 12 p.m. [37]. Following informed consent, participants were first rechecked against the exclusion criteria upon their arrival for their session. Participants then completed information on demographic factors, medication use, and perceived social support (see below). Blood (20 ccs) was then drawn and treated with EDTA to prevent clotting. Plasma was separated via centrifugation, and levels of IL-6 and CRP were determined during batch analyses at a later date (see below). Couples were then debriefed and received USD 60.00 each for their participation. All measures of demographics, medication use, and perceived social support were obtained at the time of the visit.

### 2.3. Measures

Interpersonal Support Evaluation List (ISEL). The ISEL subscales of appraisal, tangible support, and belonging were used for this study, omitting the subscale of self-esteem. This was carried out to better assess the perceived availability of global support. Cohen et al. (1985) reported that the scale’s internal consistencies were high, with a four-week test–retest reliability of 0.87. The reliability of the ISEL has also been established over a six-month period [38]. In the present study, the scale’s internal consistency was high (wives α = 0.91; husbands α = 0.93).

Inflammation Assessments. High-sensitivity CRP (hsCRP) was measured through immunonephelometry using a Behring Nephelometer II. The limit of detection for C-reactive protein was 0.015 mg/L (High Sensitivity CRP, Dade Behring). All samples were assayed in the same run, yielding a within-assay CV% of <4.5% for hsCRP. IL-6 was determined using a commercially available high-sensitivity ELISA (hsIL-6 Quantikine, R&D systems), which had a lower detection limit of 0.15 pg/mL and yielded an intra-assay CV% of <6%. Consistent with prior work, CRP and IL-6 were natural log transformed to normalize the distribution prior to analyses [39]. Significant Shapiro–Wilk tests confirmed the non-normality of the data for CRP (*W* = 0.67, *p* < 0.001) and IL-6 (*W* = 0.83, *p* < 0.001), which are among the most powerful tests of normality based on Monte Carlo simulations [40]. To create an overall index of inflammation (and reduce the number of statistical tests), CRP and IL-6 were standardized and then averaged. Similar to prior work, these separate indices were correlated at 0.48 (wives) and 0.60 (husbands).

### 2.4. Data Analysis

PROC MIXED (SAS Institute, Cary City, NC, USA) was utilized to examine actor–partner perceived support influences on inflammation following the recommendations of Campbell and Kashy [41]. Standard variables including age, gender, body mass, and medications linked to chronic conditions were statistically adjusted for [37,42,43]. The covariance structure for the dyad’s repeated-measures factors (i.e., husband, wife) was modeled using the compound symmetry structure [41]. The resulting actor–partner models allowed us to test if one’s own levels of perceived support (actor influences) and a partner’s level of perceived support were significantly related to one’s own outcomes [44].

## 3. Results

### 3.1. Descriptive Analyses

Table 1 lists the descriptive information on our primary measures of social support and inflammation. Overall perceived support levels were above the scale midpoint of 2.39 on a 0 to 4-point scale. Mean levels of inflammation were consistent with a healthy sample. Table 2 contains the raw correlation matrix for our main measures for wives and husbands separately, with the cross-diagonals indicating within-couple correlations. The use of statins and anti-inflammatory drugs was controlled for during the analyses shown in Table 2.

### 3.2. Actor and Partner Perceived Support Influences on Inflammation

Prior work has revealed relatively small effect sizes between one’s own perceived social support and inflammation [11]. Similar to this prior research, although the levels of one’s own perceived support (actor influences) were negatively related to inflammation, these links were not significant (*p* = 0.19). However, a spouse’s level of perceived support was related to lower levels of one’s own inflammation (*b* = −0.32, *SE* = 0.14, *p* = 0.02), as seen in Table 3 below. Ancillary analyses on hs-CRP and IL-6 separately revealed the same pattern of partner influences on both these measures of inflammation.

### 3.3. Ancillary Analyses

Several ancillary analyses were conducted to rule out alternative explanations. First, statistical controls for health behaviors, including weekly exercise, smoking, and alcohol consumption, were covaried during main model analyses. All main results remained unchanged. Due to socioeconomic status being a reliable predictor of inflammation [45], analyses that statistically adjusted for family income were completed, and all the main results remained unchanged.

## 4. Discussion

The main goal of this study was to test actor–partner models linking perceived social support to inflammation. The main results indicate that a partner’s level of perceived support was related to lower inflammation in their spouse. These results remained unchanged while statistically controlling for a number of demographic variables, including medication use and one’s own level of perceived support. These data highlight the importance of taking a dyadic perspective on perceived support, given the interdependence of individuals within relationships.

It was predicted that both actor and partner levels of perceived support would be related to lower levels of inflammation. Only partner levels of perceived support were associated with lower inflammation. There are a number of plausible reasons for the existence of such an association, and it requires future research. Given that marital relationships are among the most important in adulthood, a spouse with higher perceptions of support might be associated with smoother interpersonal interactions in daily life and over time. Importantly, such processes might not be captured by actor measures of support, which only reflect an individual’s own perception of support and their own interpersonal functioning. For instance, prior work indicated that social support is related to better social skills [46]. It is thus possible that partners high in perceived support may engage in more effective and responsive support, such that actors benefit from its positive influence [15,47]. Unfortunately, this study did not assess more specific interpersonal skills, and this requires future research. More generally, these explanations highlight the importance of a broader consideration of mechanisms that could be explored in future research.

It is also possible that this study captures the interpersonal and intrapersonal synchrony of perceived support within a relationship. At least one study has linked interpersonal mechanisms and close relationships to health [48] through the coregulation of emotion, cognition, behavior, and physiology within the dyad. This coregulation or psychophysiological homeostasis between partners has been seen to affect physical and emotional health [49]. Nonetheless, the interplay between intrapersonal and interpersonal perceptions within the dyad would need to be addressed in future research.

Although the directionality of effects cannot be investigated within cross-sectional data, there is some evidence suggesting a coregulation of inflammation and social behavior that could be an interesting avenue for future work. Two reviews on inflammation and social processes found that inflammatory processes regulated social behavior, inasmuch that neural sensitivity to positive and negative social feedback was seen to be enhanced [50,51]. Therefore, inflammation levels may have contributed to social behaviors that elicited support from partners. That is, when the need for support is recognized and perceived by the partner, it has the potential to influence actor levels of inflammation. Although this is speculative and requires future work, it is consistent with the growing literature documenting the bi-directional links between inflammation and social processes.

Significant relationships were found between actor–partner perceived social support, inflammation, and BMI, and between actor–partner perceived social support, inflammation, and gender. Research shows that there is a substantially increased risk of obesity when one’s partner is obese. It is thought that relationships can normalize the idea of obesity, making it more acceptable to become obese [52]. Additionally, studies have shown that increased obesity can then lead to an increase in inflammation [53,54]. In sum, the normalization of a higher BMI through one’s partner increases the likelihood of an increased actor BMI and the associated inflammation seen with increased adipose tissue. Regarding the relationship seen with gender, inflammation was seen to be lower in women than in men. This is thought to be due to the general anti-inflammatory properties associated with estrogen [55,56].

There are many limitations that should be raised. Although general perceived support has been linked to marital functioning [57,58], we need to be appropriately cautious in interpreting these results as reflecting specific interpersonal transactions within the context of marriage. As ancillary actor–partner analyses revealed that there were no significant partner effects on marital quality [59] or perceived stress [60], the precise mechanisms responsible for such a link need further exploration (e.g., social skills). This study was also cross-sectional, meaning causal inferences cannot be made. Future studies should include longitudinal designs or daily diary studies that examine support processes and health over short periods of time. In addition, our sample size, although moderate for inflammatory studies, was probably not large enough to detect actor influences. Furthermore, even though an inflammatory index was created from measures of CRP and IL-6, having just one assessment does not capture fluctuations that occur over time. Although inflammation is related to future health problems, other biological mechanisms would have bolstered our assessments. For instance, measures of cardiovascular reactivity, ambulatory blood pressure, and/or cellular aging might have improved the generalizability to health outcomes [61,62,63]. At least one study showed that perceived support influences other measures such as ambulatory blood pressure [24]. Nonetheless, none of the studies referenced above modeled actor–partner effects. Lastly, our findings may only be generalizable to populations with demographics similar to our own. These limitations notwithstanding, this is one of the first studies linking partner levels of support to a biological mechanism related to health. Hence, these results are novel and expand our thinking on antecedent processes and mechanisms linking perceived support to health that can be examined in future work.

## 5. Conclusions

Contrary to our predictions, actor levels of perceived support did not predict lower levels of inflammation. However, in the expected direction, the association was not significant. Based on a recent meta-analysis, the effect size linking social support to inflammation is small, at r = −0.07 [11]. Hence, it is likely that the current study was underpowered to detect such an association at a conventional level of significance. The nonsignificant finding for actor support also highlights the relatively larger effect sizes for partner levels of support. If similar results are confirmed in future studies, they may underscore the importance of dyadic approaches to psychosocial risk, even for psychosocial risk factors that have traditionally been conceptualized as intrapersonal in nature.

## Figures and Tables

**Table 1 ijerph-19-00799-t001:** Mean and standard deviation (SD) for main study variables.

Variable	Mean (Range)	SD
Perceived Support	2.39 (1–4)	0.39
Interleukin-6 (pg/mL)	1.57	1.05
C-reactive protein (mg/L)	0.19 (0–10)	0.25
Body mass (kg/m^2^)	26.25 (18.5–40)	4.61
Age (years)	56.2	7.30
	Frequency	
Ethnicity (% white)	94.6%	
College educated	70.3%	
Annual income over USD 40,000 per year	87.6%	
Statin use	7.75%	
Anti-inflammatory use	25.75%	

Note. *n* = 94.

**Table 2 ijerph-19-00799-t002:** Zero-order correlations among main study variables for women (top panel) and men (bottom panel), with diagonals representing cross-spouse correlations.

Variable	1. ISEL	2. BMI	3. Age	4. Statin	5. Anti-Inflam.	6. HRT	7. Inflam. Ind
1. ISEL	**0.04**	−0.05	0.06	0.06	−0.14	−0.06	0.02
2. BMI	−0.16	**0.35 ****	−0.05	0.10	0.16	−0.04	0.58 **
3. Age	−0.02	−0.02	**0.90 ****	0.20	−0.20	0.10	−0.06
4. Statin	−0.01	0.00	0.25 *	**0.34 ****	−0.02	0.09	0.05
5. Anti-Inflam.	−0.05	0.02	−0.09	0.08	**0.20**	0.01	0.09
6. HRT	-	-	-	-	-	**-**	0.06
7. Inflam. Index	−0.20	0.42 ***	0.05	0.06	0.01	-	**0.05**

Note. BMI = body mass index, Inflam. = inflammatory, HRT = hormone replacement therapy, * *p* < 0.05, ** *p* < 0.01, *** *p* < 0.001. *n* = 88 for women, 91 for men.

**Table 3 ijerph-19-00799-t003:** Actor and partner perceived support and inflammation.

Variable	Inflammatory Index		
	*b*	*S.E.*	*p*
Body Mass	0.09	0.01	0.00 **
Age	0.00	0.01	0.87
Gender (male–female)	0.24	0.12	0.05 *
Statin	0.09	0.22	0.69
Anti-inflammatory	0.03	0.12	0.80
Hormone replacement	0.12	0.24	0.50
Actor ISEL	−0.05	0.14	0.72
Partner ISEL	−0.32	0.14	0.02 *

* *p* < 0.05; ** *p* < 0.01; *n* = 179.

## Data Availability

All data can be provided upon request.

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
