# Peer review of "Partner’s Perceived Social Support Influences Their Spouse’s Inflammation: An Actor–Partner Analysis"

_ijerph, 2022, doi:10.3390/ijerph19020799_

Round 1

Reviewer 1 Report

In this study, Dr. Landvatter and colleagues investigated the association between perceived social support (PSS) and inflammation. Overall, the manuscript could be insightful. However, some issues need clarifications:

  • I am confused by the study design. Why don't the authors divide the samples into two groups with high PSS and low PSS, then assess the inflammatory markers? OR perhaps the authors could assess the individual data with regression analysis (and show the scatter diagrams)? At present, I don't really know whether the findings are interpretable and meaningful. 
  • Inflammation is a dynamic variable that can change acutely (in minutes to hours), while PSS is a more long-term / persistent approach. I think it is very important to justify the relevance of this study and how it can be applied in real life. 
  • Line 90: how to assess the "relatively healthy" status mentioned there? Please specify in the methods.
  • Older adults poses a risk for cardiovascular diseases, particularly ischemic heart diseases that could increase the level of hsCRP. Diabetes is also known to be pro-inflammatory. What did the authors do to make sure that the results are not confounded by comorbidities of the samples? 
  • What about old people with arthritis who consume NSAIDs regularly? I don't see that this factor is included in the exclusion criteria. Also, in Table 1, we can see that 1/4 of the samples used anti-inflammatory agents. How to make sure that they didn't confound the results?
  • Please provide a list (table) for the medications used by the included individuals. The authors could classify them into groups. At present, only statin was mentioned and I am quite sure that they use more medications than just a statin. 
  • Line 97: "Medication use was coded dichotomously based..." what does it mean by "dichotomously"? So only Yes/No?
  • Table 1 needs to be improved. What is the meaning of PSS of 2.39? Is it high or low? and because there is no other group to compare, I am not sure how to interpret this number. At least provide a cut-off value next to it. 
  • Please also add the cut-off value for IL-6, CRP and BMI in Table 1. 
  • I probably miss this one but what are those 1-7 in Table 2? Are they the main study variables for women? This table should be improved. it is unclear. Add a label showing which one is men and which one is women.
  • Line 12: should be "ninety-four"
  • Line 89: "All methods and procedures within this study were pre-approved by an Institutional Review Board" Which IRB approved it? Please clarify.
  • The authors need to make sure that they explain all the numbers reported in the tables. For example, Table 2 was not explained in the text at all so my assumption is that this table is not important? If so, please remove it from the manuscript.
  • Also, it seems that there is no correlation between ISEL and inflammatory index in Table 2. This aspect needs to be discussed since it seems contradictory to the claim made by the authors that "partner’s level of perceived support was related to lower inflammation in their spouse".
  • Please remove text in lines 22-23.
  • Table 1: what is the use of the word "frequency" in the middle of the table?
  • Limitations of the study should be placed before conclusion. Consider moving it to the last part of the discussion.

Author Response

Jan 3, 2022  International Journal of Environmental Research and Public Health

Dear Reviewer 1:

        I am writing in regard to our manuscript (ijerph-1520679) titled “Partner’s Perceived Social Support influences their Spouse’s Inflammation: An Actor-Partner Analysis” coauthored with Drs. Bert Uchino, Timothy Smith and Jos Bosch.  We thank you for reviewing the manuscript and are very pleased you find the manuscript acceptable for publication.

    1. I am confused by the study design. Why don't the authors divide the samples into two groups with high PSS and low PSS, then assess the inflammatory markers? OR perhaps the authors could assess the individual data with regression analysis (and show the scatter diagrams)? At present, I don't really know whether the findings are interpretable and meaningful. 

Our Response: The studies main goal was to investigate actor-partner models linking perceived support to inflammation and to explore the interpersonal components of perceived support. Due to the inherent analytic challenges of dyadic data, the Actor-Partner Interdependence Model (APIM) is recommended (Kashy & Kenny, 2000) for questions regarding the interdependence between dyad members. This interdependence between dyad members is seen to be most accurately captured using hierarchical linear modeling (HLM) and PROC MIXED models, due to its better fit for mixed effect models (Campbell & Kashy, 2002). In addition, proc mixed is a random coefficient regression model which allows one to examine continuous predictors instead of dichotomizing variables which decreases power and sensitivity. We hope this resolves your concerns regarding the research designs of this study.

    2. Inflammation is a dynamic variable that can change acutely (in minutes to hours), while PSS is a more long-term / persistent approach. I think it is very important to justify the relevance of this study and how it can be applied in real life. 

Our Response: The reviewer is correct in assuming that inflammation is a dynamic variable that can change acutely. That being said, CRP and IL-6 are considered to be inflammatory markers that favor chronic inflammatory pathways (Luan & Yao, 2018; Gabay, 2006). Furthermore, CRP and IL-6 are common measures of inflammation obtained within many clinical settings. The dynamic nature of inflammation is now addressed within the limitations of the manuscript.

    3. How to assess the “relatively healthy” status mentioned there? Please specify in the methods.

Our Response: Similar to our response to question four, chronic conditions were screened for in our study. This information is now included in the methods section.

     4. Older adults pose a risk for cardiovascular diseases, particularly ischemic heart diseases that could increase the level of hsCRP. Diabetes is also known to be pro-inflammatory. What did the authors do to make sure that the results are not confounded by comorbidities of the samples? 

Our Response:  Conditions related to cardiovascular disease were screened for, including diabetes. This is now clarified within the methods section. We also assessed health behaviors and relied on CRP values (>10) to determine if acute illnesses were confounding the inflammatory assessments.  No participant had a CRP value over or close to 10. This is now clarified within the methods section.

     5.  What about old people with arthritis who consume NSAIDs regularly? I don't see that this factor is included in the exclusion criteria. Also, in Table 1, we can see that 1/4 of the samples used anti-inflammatory agents. How to make sure that they didn't confound the results?

Our Response: The use of NSAID’s were controlled for in our analyses shown in table 2. This is now explicitly mentioned within the results section.

     6. Please provide a list (table) for the medications used by the included individuals. The authors could classify them into groups. At present, only statin was mentioned, and I am quite sure that they use more medications than just a statin. 

Our Response:  The following is a list of medications that were coded for within our study: thyroid, statins, reflux, allergies, osteoporosis, hormone replacement therapy, inflammation, hypertensive, other cardiovascular, and diabetes medications.

  1. Medication use was coded dichotomously. What does it mean by dichotomously?

 Our Response: Yes, you are correct. Participants were given only yes or no options for answers. This is now included within the methods section.

  1. Table 1 needs to be improved. What is the meaning of PSS of 2.39? Is it high or low? and because there is no other group to compare, I am not sure how to interpret this number. At least provide a cut-off value next to it. Please also add the cut-off value for IL-6, CRP, and BMI in table1.

Our Response: The ranges for perceived support, CRP, and BMI have now been included within the table. A medical cut-off for IL-6 still has not been established, although a recent meta-analysis has found a range from 0-43.5 to be found within healthy populations (Said et al., 2020) https://pubmed.ncbi.nlm.nih.gov/33155686/

    9. Line 12 should be “ninety-four”

Our Response: Thank you for catching this error. This has now been corrected within the abstract.

  1. Please remove text in lines 22-23.

Our Response: The title from the original upload of the manuscript has now been removed from lines 22-23.

  1. Table 1: what is the use of the word “frequency” in the middle of the table?

Our Response: Frequency was used to indicate the change from means to percentiles within the same column. Frequency has now been placed in a more logical location to indicate this change.

    12. Limitations of the study should be placed before conclusion.

Our Response: Limitations have now been placed before the conclusion.

  1. All methods and procedures within this study were pre-approved by an Institutional Review Board”. Which IRB approved it?

Our Response: The study was pre-approved by the Institutional Review Board at the University of Utah. This information has now been included within the methods.

    14. I probably miss this one but what are those 1-7 in Table 2? Are they the main study variables for women? This table should be improved. it is unclear. Add a label showing which one is men and which one is women.

    15. The authors need to make sure that they explain all the numbers reported in the tables. For example, Table 2 was not explained in the text at all so my assumption is that this table is not important. If so, please remove it from the manuscript.

    16. Also, it seems that there is no correlation between ISEL and inflammatory index in Table 2. This aspect needs to be discussed since it seems contradictory to the claim made by the authors that “Partner’s level of perceived support was related to lower inflammation in their spouse”.

Our Response(14-16): The numbers in the top column correspond to the numbers and labels in the left column. The diagonal of the correlation matrix is across dyads. The top labels have now been made to saliently mirror the variables in the left column. The diagonal in this table, going from 1,1, 2,2,…. to 7,7, represent the cross-spouse correlation on those specifically labeled measures. The measures above the diagonal correspond to women, while the measures below the cross-spouse correlations, correspond to men. The cross-spouse correlation is what delineates the correlation coefficients by gender. It is our hope that this clarifies any questions you had regarding Table 2 and the use of it within our manuscript. Table 2 is referred to in the descriptive analyses section of the paper.

We would again like to thank you and other reviewers. Please let us know if there is anything else needed and we hope that you now find the paper ready for publication in the International Journal of Environmental Research and Public Health.  Thank you in advance and we look forward to hearing from you.

Reviewer 2 Report

General Comment:  Thank you for the opportunity to review this interesting and well-written manuscript reporting on the relationship between inflammation and perceived social support in married couples.  A few items outlined below if addressed would strengthen this report.

Abstract:   Ninety-four misspelled line 12.  Consider clarifying couple gender.  Were they all cisgender male/female?

Intro:  Background adequate – some citations dated.  Research question clear with directional hypothesis indicated in last paragraph of this section.

Methods:  Content on participants seems appropriate for results section.  Rather see more information on when, where, and how participants were recruited.  What is the general health of the population you recruited from? You could indicate BMI since this is an outcome measured in your participants.  Under procedures – did dyads present together?  Where was data collected?  What was covered in the couple debrief?  Was BMI self-report height and weight or were they measured?  Did you measure years married?

Results:  Table 1 – is N=94 correct?  Is the table then reflecting one of the dyad? 

Discussion:   Line 188 appears to be incomplete….”related to better social….” Would like to see discussion on the other significant relationships – BMI and Gender.

Author Response

Jan 3, 2022 International Journal of Environmental Research and Public Health

Dear Reviewer 2:

        I am writing in regard to our manuscript (ijerph-1520679) titled “Partner’s Perceived Social Support influences their Spouse’s Inflammation: An Actor-Partner Analysis” coauthored with Drs. Bert Uchino, Timothy Smith and Jos Bosch.  We thank you and the other reviewers for the helpful suggestions and comments regarding our manuscript.  In the remainder of this letter, we detail how we have responded in the revision (via track changes), or here, to each of the suggestions received from your review. 

  1. Ninety-four misspelled in line 12 of the abstract. Consider clarifying couple gender.

Our Response: Thank you for catching our error and for your suggestion. Spelling has been corrected and gender clarified within the abstract.

  1. What is the general health of the population you recruited from?

Our Response: At the time of the study, no participants reported chronic disease. In addition, all participants were screened for HIV, Cancer, or any conditions related to cardiovascular disease. This has also been clarified in the methods section of the manuscript.

  1. Could you indicate BMI since this is an outcome measured in your participants?

Our Response: We do have BMI on each participant and BMI was controlled for in our analyses.

  1. Under procedures – did dyads present together?

Our Response: Yes, dyads arrived and were presented together. This is also now clarified within the procedure section.

  1. Where was the data collected?

Our Response: Data was collected in our main laboratory at the University of Utah. This information has now been included within the procedure section of the manuscript.

  1. What was covered in the couple debrief?

Our Response: At the conclusion of the study, couples were told the purpose of the study (looking at inflammatory bio-markers within dyads) and were given the opportunity to ask any questions before they left.

  1. Was BMI self-reported?

Our Response: Yes, measures of height and weight were self-reported within this study.

  1. Did you measure years married?

Our Response: Yes, the average years married was 27.3.

  1. Table 1 – is N=94 correct? Is the table then reflecting one of the dyad?

Our Response: Yes – 94 is the number of dyads as that is the unit of analysis. There were a total of 94 men and 94 women equaling 94 dyads.

  1. Line 188 appears to be incomplete.

Our Response: Thank you for catching this error. This has now corrected within the results section of the manuscript.

  1. Would like to see discussion on the other significant relationships – BMI and Gender

Our Response: These are important relationships that we have now addressed within our discussion section of the manuscript. Research shows that a person’s chance of becoming obese increases substantially when he or she has a friend, sibling, or partner who has become obese. When the social norms associated with obesity become more normalized through one’s relationships, it can make it more acceptable to become obese (Christakis & Fowler, 2007). Additionally, with increased obesity, research shows an increase in one’s inflammation (McArdle et al., 2013; Burhans et al., 2018). With the BMI of one’s partner influencing one’s own BMI, there is also a greater likelihood of inflammation associated with increased BMI. Regarding gender, inflammation was seen to be lower in women than in men. This is thought to be due to the general anti-inflammatory properties associated with estrogen (Monteiro et al., 2014; Villa et al., 2015). Importantly, this study replicates prior work linking BMI and gender to inflammation.

           We would again like to thank you and other reviewers. Please let us know if there is anything else needed and we hope that you now find the paper ready for publication in the International Journal of Environmental Research and Public Health.  Thank you in advance and we look forward to hearing from you.

Reviewer 3 Report

Congratulations on a very interesting paper. 

It is true that the sample size is probably not large enough; but considering that it is one of the first studies linking partner levels of support to a biological mechanism related to health, I think it should be accepted for publication.

Author Response

Jan 6, 2022

Dear Reviewer 3:

            I am writing in regard to our manuscript (ijerph-1520679) titled “Partner’s Perceived Social Support influences their Spouse’s Inflammation: An Actor-Partner Analysis” co-authored with Drs. Bert Uchino, Timothy Smith, and Jos Bosch. We thank you for taking the time to review the manuscript and are very pleased you find the manuscript acceptable for publication.

Round 2

Reviewer 1 Report

Thanks for the responses. I only have one minor suggestion to move the limitation to the end of the discussion. 

Author Response

Jan 4th, 2022 International Journal of Environmental Research and Public Health

Dear Reviewer 1:

I am writing in regard to our manuscript (ijerph-1520679) titled “Partner’s Perceived Social Support influences their Spouse’s Inflammation: An Actor-Partner Analysis” co-authored with Drs. Bert Uchino, Timothy Smith, and Jos Bosch. We thank you for reviewing the manuscript and hope you find the manuscript acceptable for publication.

  1. I only have one minor suggestion to move the limitation to the end of the discussion.

Our Response: Thank you for your recommendation facilitating the flow of the manuscript. Limitations have now been moved to the end of the discussion section. 

Thank you for taking the time to review our manuscript and we hope that you now find it ready for publication in the International Journal of Environmental Research and Public Health.